# Lower Limb Exoskeleton for Rehabilitation with Flexible Joints and Movement Routines Commanded by Electromyography and Baropodometry Sensors

**DOI:** 10.3390/s23115252

**Published:** 2023-06-01

**Authors:** Yukio Rosales-Luengas, Karina I. Espinosa-Espejel, Ricardo Lopéz-Gutiérrez, Sergio Salazar, Rogelio Lozano

**Affiliations:** 1Centro de Investigación y de Estudios Avanzados del Instituto Politécnico Nacional (CINVESTAV), Av. IPN #2508, San Pedro Zacatenco, Mexico City 07360, Mexico; 2Investigador por México-Consejo Nacional de Humanidades, Ciencias y Tegnologías (IXM-CONAHCYT), Av. de los Insurgentes Sur #1582, Crédito Constructor, Benito Juárez, Mexico City 03940, Mexico; 3CNRS UMR 7253 Heudiasyc, Université de Technologie de Compiegne, 60203 Compiegne, France

**Keywords:** electromyography sensors, baropodometry sensors, torque sensors, exoskeleton, human intention, classifier of signals

## Abstract

This paper presents the development of an instrumented exoskeleton with baropodometry, electromyography, and torque sensors. The six degrees of freedom (Dof) exoskeleton has a human intention detection system based on a classifier of electromyographic signals coming from four sensors placed in the muscles of the lower extremity together with baropodometric signals from four resistive load sensors placed at the front and rear parts of both feet. In addition, the exoskeleton is instrumented with four flexible actuators coupled with torque sensors. The main objective of the paper was the development of a lower limb therapy exoskeleton, articulated at hip and knees to allow the performance of three types of motion depending on the detected user’s intention: sitting to standing, standing to sitting, and standing to walking. In addition, the paper presents the development of a dynamical model and the implementation of a feedback control in the exoskeleton.

## 1. Introduction

In recent years, rehabilitation systems have been developed in order to assist patients with their movements and restore their motor ability. Exoskeletons have contributed to patients’ health by helping them to carry out physiotherapeutic routines [1]. Among the tasks with which the exoskeletons assist, we can mention support for walking, for standing, and for the execution of specific therapy routines [2,3,4]. The exoskeletons for patients or elderly care are responsible for helping people with some type of weakness to carry out their daily tasks as in [5], in which the authors developed a lower limb exoskeleton with the purpose of assisting the sit-to-stand and stand-to-sit tasks for people with spinal cord injury or elderly people. In addition, exoskeletons applied to rehabilitation have become an important tool for therapists; they reduce the risks to their health caused by the transport of people [6]. There are various architectures that can be used for bilateral training; for lower limbs, mainly for adults, as in [7], where three healthy male subjects aged 27, 25 and 24, and three subjects undergoing stroke rehabilitation, two men and one woman aged 54, 68 and 30, were enrolled in experiments with walking rehabilitation exoskeletons. Some authors place great emphasis on the mechanical and electronic design of the exoskeleton as a fundamental part of its development, as in [8,9], where the authors present in detail the mechanical design of their exoskeletons.

An important part of the research on exoskeletons consists of the actuators that will be used. In [10], many papers on exoskeletons that were developed using springs as part of their structure were reviewed, such as: series elastic actuators (SEA), parallel elastic actuators (PEA) and elastic actuators of the magneto-rheological series (MRSEA) and other interactions. According to this paper, the use of SEA actuators is important given that they ensure that the coupling between the user and the motor is compliant, thereby protecting the user’s joints from impact loads and other undesirable interactions. Furthermore, as mentioned in [10], “the compliance introduced by the spring facilitates a torque-based control strategy by transforming the torque/force control problem into a position control problem based on the measurement of the springs deformation. These sensors, widely used in exoskeletons, allow a smooth force transmission, accurate force control, lower output impedance, shock tolerance, energy efficiency, and back-drivability in human–robot physical interactions. The spring acts as an impact damper and reduces the actuator inertia felt by the user, thus allowing the user to increase safety and comfort. A further advantage is the peak motor power reduction exploiting the spring capacity of storing energy. In [11], the authors outline the hardware of two printed circuit board (PCB) designs for collecting and conditioning sensor feedback from two SEA subsystems and an inertial measurement unit (IMU).

One of the recently explored areas is the detection of the intention of motion to activate specific movement routines in exoskeletons [12,13]. The human intention (HI) is defined as the determination of a person to perform a movement, which is a fundamental component in the development of exoskeletons and is the goal of this paper. Human intention detected by the correct sensors produces a human–machine interaction in which the patient collaborates with the device during the recovery process. Electromyographic signals can be used as an alternative to detecting the human intention, which are manifestations of the movement generated 30 ms before an action is performed [14,15]. Some studies that apply surface electromyography (sEMG) use the sEMG-driven musculoskeletal (MS) model, which establishes a relationship between signals and joint moment, angular velocity or acceleration [16].

Another group of studies implements machine learning (ML). Machine learning performs discrete motion classification or continuous motion estimation by mapping sEMG inputs to the human movement intention through techniques such as neural networks (NN), vector support machines (VSM), and K-means (KM), among others [14]. In [16], the authors estimated the torque of the knee joint exerted by the set of muscles used for the extension and flexion of the knee, in real time with the implementation of a Hill-type musculoskeletal model. In [17], the authors accomplished the estimation of the torque using sEMG and the optimal control of adaptive impedance during rehabilitation assisted by an active knee orthosis. The optimization model was performed by comparing the estimated sEMG torque with the torque generated by the inverse dynamics tool of the OpenSim software. As an alternative solution, they proposed a multilayer perceptron neural network (NN) to map the EMG signals to the user’s torque. In [18], the authors presented an electric assistance robot that adapts to human intention and a method to recognize the successive phases that a subject performs during the movement of sitting and standing. Surface electromyographic signals (sEMG) and the ground reaction force (FGR) were used as inputs of the algorithm. A neural network assembly and a window of 0.1 s were used to identify each phase.

In this paper, an exoskeleton was developed with rotational elastic type actuators which are a variation of the SEA actuators, whose operation is explained in more detail in Section 3.2. The exoskeleton prototype has six degrees of freedom divided in four DoF for the lower limbs (two for knees joints and the other two for hip joints) and two DoF as part of the mechanism that allows standing and sitting exercises. Mechanically, the exoskeleton can be divided into two main parts, the lifting system, and the lower limb system. In Figure 1, we see these two parts that integrate the complete prototype. The lifting system holds the patient by the torso using a safety harness and is responsible for supporting the full weight of the patient, and is actuated by two linear motors and a 4-bar system used to sit and stand the patient. The lower limb system is an anthropomorphic exoskeleton that supports the patient’s legs, which is actuated by four motors whose axes coincide with the axis of rotation of the patient’s hip and knee joints. The mechanical design of the prototype allows the exoskeleton to take the human patient and assist him/her from sitting to standing position for later assisting him/her in a gait rehabilitation exercises.

The exoskeleton prototype also contains a human intention detection system based on electromyography; these signals along with the signals from the baropodometry sensors are classified by an algorithm that decides which routine (walking, sitting or standing) the exoskeleton should perform. These routines are already programmed into a desired trajectory suitable for each joint of the exoskeleton so that the classification system only generates a movement command.

Finally, the exoskeleton contains a feedback control system, which performs trajectory tracking. When the desired trajectory is finished, the human intention detection system waits again for an effort from the patient to classify the signals obtained and again decide which cycle to start reproducing.

To develop the control law, a dynamical model of the exoskeleton was obtained considering its flexible joints, but a dynamical model of the human is not obtained because we are performing only passive rehabilitation. That means that the patient must not make any effort during the performance of the exoskeleton movement routine. If for some reason the human generates any movement or effort, these are considered disturbances and the proposed control is capable of rejecting them.

Then, the goal of this paper was the use of a signal classifier of baropodometry, electromyography, and torque sensors as a human detection system to develop a lower limb therapy exoskeleton with a feedback control law that considers the non-linearities of the mathematical model to obtain numerical and real-time experimental results.

The paper is organized as follows. In Section 2, the design and mechanical development of the platform is presented. In Section 3, the development of the exoskeleton dynamical model with elastic joints and the obtaining trajectories of motion are presented. The electromyography and baropodometry sensors are described in Section 4. The design and development of the control law are presented in Section 5. The numerical results and real-time experimental results are presented in Section 6. Finally, the conclusions are presented in Section 7.

## 2. Platform Development

The developed lower limb exoskeleton was specifically designed to meet the following requirements:Adjust the sizes of the Latin American population;Support the patient’s weight without the need for a crane;Assist in the task of retraining training, as well as in the task of sitting and getting up.

In addition, the design must be ergonomic and easy to place.

The final design consists of three parts: the exoskeleton legs with actuators and sensors, the exoskeleton lifting system, and the control unit. The exoskeleton rehabilitation movements are made through a series of links attached by joints that correspond to the patient’s knee and hip joints.

### 2.1. Exoskeleton Lower Limb System

The exoskeleton’s legs, shown in Figure 2, were designed with an anthropomorphic approach to allow easy adjustment of the patient’s legs. In this sense, the exoskeleton was specifically designed so that it could fit the patient in front of the body and not in the back, with the intention that the patient can be adjusted while sitting, and help the patient to get up and then start the rehabilitation exercises. In addition, from the sitting position, the exoskeleton can help in exercises such as knee extension or the process of sitting and getting up. Considering the variations in population measures, an extension system was added in the links of the legs and trunk, with the purpose of covering a slightly wider range of patients.

The required torque for each joint was provided by an AHC2 family Bosh CD engine, through an elastic transmission, which generates up to 20 Nm of torque. Two absolute encoders model AMT20 of the company CUI INC; Tualatin, U.S.A., with a resolution of 12 bits and SPI interface, were used for each joint to measure the angular position of both the link and the motor, and the angular speeds and accelerations can be calculated.

In order to keep the user safe, we should avoid hyperextension in the joints. Mechanical stops are added in all the joints to obtain the range of motion (RoM) described in Table 1.

### 2.2. Exoskeleton Lifting System

The lifting system of the exoskeleton is responsible for carrying the entire weight of the patient, from the sitting position to the standing position and for the entire rehabilitation session, leaving the rehabilitation task to the legs only. The exoskeleton lifting system is made up of a double four-bar system operated by two linear actuators, model LACT6P, which can withstand a load of 50 kg each, see Figure 3. It is anchored, at one end, to the hand supports of a standard treadmill and, at the other end, joins the mechanical legs, allowing them to go up and down while always maintaining its vertical.

The patient is supported by a harness (not shown) that is attached to the upper bar of the harness frame, which is joined to the four-bar system in the same place where the mechanical legs are connected. Thus the exoskeleton is able to comfortably lift patients weighing more than average.

### 2.3. Control System

The control system consists of the real-time control electronic card and the host computer with the interface for the physiotherapist.

The control electronic card, manufactured by National Instruments© (Austin, TX, USA), is responsible for running the controller in real-time, receiving data from absolute encoders and analog sensors, and operating the motors through PWM outputs which are connected to H-bridge controllers; it communicates with the host computer via a USB connection, and also has an emergency stop system available for the patient. It has two secondary cards designed ad hoc for the connection of all the sensors and motors of the exoskeleton, which has six degrees of freedom.

The main computer runs the physiotherapist’s graphic interface, allowing him to command all the movements and exercises that the exoskeleton can execute. For development stages, the main computer is used to program new control laws and test their performance; in this case, the interface shows graphs of all the variables of interest as well as the desired trajectories

## 3. Mathematical Model

In this section, we present the mathematical models and the desired trajectories for the exoskeleton.

### 3.1. Dynamical Model of the Exoskeleton

Initially, the dynamical model of the exoskeleton was obtained by considering the simplified model with rigid joints as shown in Figure 4, in which there were two links corresponding to the thigh (link 1) and the leg (link 2) with lengths $l_{1}$ and $l_{2}$, and with masses $m_{1}$ and $m_{2}$, respectively.

The joints are rotational and correspond to the joint of the hip $q_{1}\left(t\right)$ and the knee $q_{2}\left(t\right)$, both form the vector of joint positions q(t) ∈ $R^{2}$.

The dynamical equations of a limb of the exoskeleton in matrix form are:(1)$$\begin{array}{cc} \; & \tau =M\left(q\right)\ddot{q}+C\left(q,\dot{q}\right)\dot{q}+g\left(q\right)\\ & \left[\begin{array}{c} \tau_{1}\\ \tau_{2} \end{array}\right]=\left[\begin{array}{cc} M_{11}\left(q\right) & M_{12}\left(q\right)\\ M_{21}\left(q\right) & M_{22}\left(q\right) \end{array}\right]\left[\begin{array}{c} {\ddot{q}}_{1}\\ {\ddot{q}}_{2} \end{array}\right]\\ & \;\;+\left[\begin{array}{cc} C_{11}\left(q,\dot{q}\right) & C_{12}\left(q,\dot{q}\right)\\ C_{21}\left(q,\dot{q}\right) & C_{22}\left(q,\dot{q}\right) \end{array}\right]\left[\begin{array}{c} {\dot{q}}_{1}\\ {\dot{q}}_{2} \end{array}\right]+\left[\begin{array}{c} g_{1}\left(q\right)\\ g_{2}\left(q\right), \end{array}\right] \end{array}$$
where *M* ∈ $R^{2\times 2}$ is the inertia matrix, *C* ∈ $R^{2\times 2}$ is the Coriolis matrix, g ∈ $R^{2}$ is the vector of gravitational forces and τ ∈ $R^{n}$ are the forces of the actuators. Furthermore, the elements of the *M* and *C* matrixes are:(2)$$\begin{array}{cc} \; & M_{11}\left(q\right)=I_{1}+I_{2}+l_{1}^{2}m_{2}+l_{c1}^{2}m_{1}+l_{c2}^{2}m_{2}+2l_{1}l_{c2}m_{2}cos\left(q_{2}\right)\\ & M_{12}\left(q\right)=I_{2}+l_{c2}^{2}m_{2}+l_{c2}^{2}m_{2}+l_{1}l_{c2}m_{2}cos\left(q_{2}\right)\\ & M_{21}\left(q\right)=I_{2}+l_{c2}^{2}m_{2}+l_{c2}^{2}m_{2}+l_{1}l_{c2}m_{2}cos\left(q_{2}\right)\\ & M_{22}\left(q\right)=I_{2}+m_{2}l_{c2}^{2}\\ & C_{11}\left(q,\dot{q}\right)=-m_{2}l_{1}l_{c2}sin\left(q_{2}\right){\dot{q}}_{2}\\ & C_{12}\left(q,\dot{q}\right)=-m_{2}l_{1}l_{c2}sin\left(q_{2}\right)\left[{\dot{q}}_{1}+{\dot{q}}_{2}\right]\\ & C_{21}\left(q,\dot{q}\right)=m_{2}l_{1}l_{c2}sin\left(q2\right){\dot{q}}_{1}\\ & C_{22}\left(q,\dot{q}\right)=0\\ & g_{1}\left(q\right)=\left[m_{1}l_{c1}+m_{2}l_{1}\right]gsin\left(q_{1}\right)+m_{2}gl_{c2}sin\left(q_{1}+q_{2}\right)\\ & g_{2}\left(q\right)=m_{2}gl_{c2}sin\left(q_{1}+q_{2}\right). \end{array}$$

### 3.2. Dynamical Model of Elastic Rotational Actuators

The exoskeleton has elastic rotational actuators. This type of actuator has a spring between the motor shaft and the mechanical joint that adheres to the person. The springs ensure that the coupling between the user and the motor is compliant, thereby protecting the users body from impact loads and other undesirable interactions.

Since the actuators transmit the movement through gears that are not totally rigid, modeling was performed with elastic joints.

An elastic joint is represented by the motor rotor (in yellow) and the link load (in blue). The angular position of the motor rotor is denoted by $q_{m}$ ∈ $R^{2}$ and the angular position of the link in front of the elastic gear by $q_{e}$ ∈ $R^{2}$ as shown in Figure 5.

The generalized coordinates are $\left[q_{e}^{T}\;q_{m}^{T}\right]$, where $q_{e}={\left[q_{e1}\;q_{e2}\right]}^{T}$ are the angular positions of links 1 and 2, and $q_{m}={\left[q_{m1}\;q_{m2}\right]}^{T}$ are the angular positions of motors 1 and 2.

The kinetic energy considering elastic joints is the sum of the kinetic energy of the links and of the rotors:(3)$$k\left(q_{e},{\dot{q}}_{e},{\dot{q}}_{m}\right)=\frac{1}{2}{\dot{q}}_{e}^{T}M\left(q_{e}\right){\dot{q}}_{e}+\frac{1}{2}{\dot{q}}_{m}^{T}J{\dot{q}}_{m},$$
where $M(q_{e})$ is the inertia matrix of the rigid robot (considering an infinite stiffness value $k_{i}\;\forall \;i$), and *J* is a positive definite diagonal matrix with values in its diagonal equal to the product of the inertia moments of the rotors and the square of the gear ratio: $J=diag\{J_{1}r_{1,1}^{2},J_{2}r_{1,2}^{2}\}$.

The potential energy is the sum of the gravitational energy stored in the torsional springs:(4)$$U\left(q_{e},q_{m}\right)=U_{1}\left(q_{e}\right)+\frac{1}{2}{\left[q_{e}-q_{m}\right]}^{T}K\left[q_{e}-q_{m}\right],$$
where $U_{1}\left(q_{e}\right)$ is the energy due to gravity considering the robot with rigid joints. Matrix *K* is the positive definite diagonal with the torsion constants on the diagonal defined as: $K=diag\{k_{1},k_{2}\}$.

Finally, using the Euler–Lagrange formalism and considering a viscous friction in the motors, it follows that:(5)$$\begin{array}{cc} \; & M\left(q_{e}\right){\ddot{q}}_{e}+C\left(q_{e},{\dot{q}}_{e}\right){\dot{q}}_{e}+g\left(q_{e}\right)+K\left[q_{e}-q_{m}\right]=0\\ & J{\ddot{q}}_{m}+B{\dot{q}}_{m}-K\left[q_{e}-q_{m}\right]=\tau , \end{array}$$
where *B* is a positive definite diagonal matrix and its values are the viscous friction parameters of each motor $B=diag\{b_{1},b_{2}\}$.

### 3.3. Trajectories of Motion

The motion trajectories of each joint were obtained by video analysis of the three movement routines: sitting to standing (STS), standing to sitting (STA), and standing to walking (SW). Markers were placed on the joints to obtain the angular positions. Figure 6 shows the images of the four phases of the cycle from sitting to standing (A-Start, B-Incline, C-Elevation and D-Stabilization) and their respective markers. The angular positions were obtained with respect to the hip and the flexion and extension values were obtained using the coordinates of the markers.

The angular trajectories were generated by an approximation of trigonometric polynomials given by:(6)$$q_{di}\left(t\right)=\frac{a_{0}}{2}+\sum_{k=1}^{n}\left(a_{ik}cos\left(k\omega_{i}t\right)+b_{k}sin\left(\omega_{i}kt\right)\right),\;i=1,2,$$
where $q_{di}$ is the path of the *i*-th joint, $\omega_{i}$ is the angular frequency of the *i*th joint, and *n* is the degree of the polynomial. The angular frequency is defined as $\omega_{i}=2\pi /T_{i}$, with period $T_{i}$ ∈ $R^{+}$. The desired trajectories corresponding to the hip and knee joint are denoted by $q_{d1}$ and $q_{d2}$, respectively.

The angular trajectories obtained for the routines sitting to standing, standing to sitting, and standing to walking, denoted by $q_{d}\mathrm{STS}$ ∈ $R^{2}$, $q_{d}\mathrm{STA}$ ∈ $R^{2}$, and $q_{d}\mathrm{SW}$ ∈ $R^{2}$, respectively, can be found below. A period $T_{i}=\pi$ is considered so that $\omega_{i}=1$ for i=1,2 (the period $T_{i}$ can be chosen appropriately) and the order of the polynomial is chosen as n=8. Figure 7 shows the desired trajectories obtained from Equations (Equation 7)–(Equation 12) which are smooth and bounded since the positions and velocities are bounded. The upper bounds are shown in Table 2.

Trajectories from sitting to standing:(7)$$\begin{array}{cc} \; & q_{d1STS}\left(t\right)=-22.35-3.66cos\left(1\omega t\right)-0.68cos\left(2\omega t\right)-0.32cos\left(3\omega t\right)-0.26cos\left(4\omega t\right)\\ & -0.13cos(5\omega t)-0.26cos(6\omega t)-0.09cos(7\omega t)-0.27cos(8\omega t)-19.31sin(1\omega t)\\ & -8.77sin(2\omega t)-5.64sin(3\omega t)-3.9sin(4\omega t)-3.39sin(5\omega t)-2.65sin(6\omega t)\\ & -2.29sin(7\omega t)-1.98sin(8\omega t) \end{array}$$
(8)$$\begin{array}{cc} \; & q_{d2STS}\left(t\right)=-34.48-3.17cos\left(1\omega t\right)-0.36cos\left(2\omega t\right)-0.52cos\left(3\omega t\right)-0.47cos\left(4\omega t\right)\\ & -0.23cos(5\omega t)-0.67cos(6\omega t)+0.13cos(7\omega t)-0.5cos(8\omega t)-29.08sin(1\omega t)\\ & -12.31sin(2\omega t)-7.62sin(3\omega t)-5.34sin(4\omega t)-4.69sin(5\omega t)-3.8sin(6\omega t)\\ & -3.2sin(7\omega t)-2.9sin(8\omega t) \end{array}$$

Trajectories from standing to sitting:(9)$$\begin{array}{cc} \; & q_{d1STA}\left(t\right)=-22.35-3.05cos\left(1\omega t\right)-0.12cos\left(2\omega t\right)+0.21cos\left(3\omega t\right)+0.23cos\left(4\omega t\right)\\ & +0.41cos(5\omega t)+0.24cos(6\omega t)+0.41cos(7\omega t)+0.23cos(8\omega t)+19.42sin(1\omega t)\\ & +8.79sin(2\omega t)+5.64sin(3\omega t)+3.9sin(4\omega t)+3.37sin(5\omega t)+2.65sin(6\omega t)\\ & +2.25sin(7\omega t)+1.99sin(8\omega t) \end{array}$$
(10)$$\begin{array}{cc} \; & q_{d2STA}\left(t\right)=-34.48-2.26cos\left(1\omega t\right)+0.42cos\left(2\omega t\right)+0.2cos\left(3\omega t\right)+0.2cos\left(4\omega t\right)\\ & +0.51cos(5\omega t)+0.05cos(6\omega t)+0.83cos(7\omega t)+0.24cos(8\omega t)+29.16sin(1\omega t)\\ & +12.31sin(2\omega t)+7.64sin(3\omega t)+5.36sin(4\omega t)+4.66sin(5\omega t)+3.86sin(6\omega t)\\ & +3.1sin(7\omega t)+2.94sin(8\omega t) \end{array}$$

Trajectories from standing to walking:(11)$$\begin{array}{cc} \; & q_{d1SW}\left(t\right)=-8.29+21.19cos\left(1\omega t\right)+0.83cos\left(2\omega t\right)-1.01cos\left(3\omega t\right)+0.01cos\left(4\omega t\right)\\ & +0.25cos(5\omega t)+0.24cos(6\omega t)-0.35cos(7\omega t)+0.05cos(8\omega t)-8.43sin(1\omega t)\\ & +6.89sin(2\omega t)+1.95sin(3\omega t)-1.07sin(4\omega t)+0.8sin(5\omega t)+0.77sin(6\omega t)\\ & +0.6sin(7\omega t)-0.22sin(8\omega t) \end{array}$$
(12)$$\begin{array}{cc} \; & q_{d2SW}\left(t\right)=-26.575-3.24cos\left(1\omega t\right)+14.12cos\left(2\omega t\right)-2.1cos\left(3\omega t\right)-1.02cos\left(4\omega t\right)\\ & -0.06cos(5\omega t)+0.68cos(6\omega t)-0.27cos(7\omega t)+0.07cos(8\omega t)-30.54sin(1\omega t)\\ & +3.13sin(2\omega t)+9.27sin(3\omega t)-0.99sin(4\omega t)+1.43sin(5\omega t)+1.27sin(6\omega t)\\ & +1.12sin(7\omega t)+0.59sin(8\omega t) \end{array}$$

## 4. Detection of Human Intention

The human intention can be detected using the flexible part of the robot joints. When the user slightly flexes the hip or a knee, the elastic element of the exoskeleton joint contracts, generating feedback to the exoskeleton control that sends a signal to move the prototype motors. This is the simplest way to control the exoskeleton but it requires the user to have at least a small degree of mobility in the joints.

When the user has some impediment to mobilize but still generates electromyography signals, the human intention can be detected with a system that interprets these signals and generates commands to perform desired tasks.

The detection system proposed and developed in this paper has two inputs corresponding to surface electromyography and baropodometry sensors that are read by the myRIO microprocessor where the signals are processed. The detection system recognizes three different movements and then generates the corresponding movement routine for the exoskeleton, as shown in Figure 8.

The electromyographic acquisition is divided into four stages: the skin preparation, the electrode position, the analog conditioning, and the analog-digital conversion. The human skin preparation consists of cleaning the skin to provide sEMG recordings with low noise levels. It ensures the removal of body hair, oils, and flaky skin layers and consequently reduces the impedance at the electrode–gel–skin interface. Using an abrasive solution and wetting clean skin with water reduces the impedance of the skin and electrodes. It also minimizes allergic responses [20].

The surface electrodes are made of silver/silver chloride (Ag/AgCl), which are transducers of the sEMG signal. The Covidien type H124SG Ag/AgCl electrode with a 24 mm diameter is used because this satisfies the requirements of the European Concerted Action Surface EMG for noninvasive assessment of muscle [21].

The assembly of the electrodes in a bipolar configuration is located between the zone of innervation and the regions of the tendon [22]. The Myoware sensor electrodes have a distance between electrodes of 30 mm, which are positioned in the middle of the muscle body aligned with the orientation of the muscle fibers on the four muscles: Rectus femoris (RF), Biceps femoris (BF), Tibialis Anterior (TA) and Gastrocnemius (GAS).

The Myoware sensor performs the assignment of conditioning the sEMG signal and the sensor stages are: the input to the AD8226 instrumentation amplifier, rectification, smoothing, and signal amplification.

The sEMG signal is acquired with the myRIO Digital Analogue converter by using the Nyquist–Shanon sampling theorem and a maximum frequency $f_{max}$ of the electromyographic signal of 500 Hz, the sampling frequency must be at least $f_{s}\geq 2f_{max}=1000$ Hz [23]. Then, the used sampling frequency is of 1 kHz. The signals are filtered using a band-pass filter type butter-worth of fourth order from 10 to 250 Hz to have better information.

### 4.1. Human Intention

The identification system detects the human intention to execute three different movements. Those movements are getting up from a chair, sitting in a chair and starting walking. The two possible initial positions are sitting or relaxed standing. The system identifies the initial position and then differentiates between getting up from a chair and the other movements with the initial standing position. Being in the relaxed standing position, distinguishes between sitting and walking intentions.


*Characteristics of the movements:*


The characteristics of the three movements are described with the activation of the muscles and the ground reaction forces in the front and rear of each foot. $F_{0}$ is the reaction force of both feet in the starting position. $F_{1}$ is the initial force to move for both feet. $F_{i}$ represents the reaction force at the left limb and $F_{d}$ the reaction force at the right limb.


*Getting up from a chair:*


In this movement of getting up from a chair, the rectus femoris muscle is an extensor of the knee that contracts and the gastrocnemius muscle contracts to perform the plantar flexion. The ground reaction force behaves like the initial value of $F_{0}$ increases to $F_{1}$ being higher at the back, generally $F_{1}>F_{0}$, which can be seen in Figure 9A.


*Sitting on a chair:*


In the initial movement of sitting on a chair, with respect to the muscles, to flex the hip the rectus femoris contracts and to flex the knee the biceps femoris contracts. With respect to the ground reaction force, the initial value $F_{0}$ increases at the back of both feet $F_{1p}$ and at the bottom of both feet the force $F_{1a}$ decreases, being the sum equal to the initial reaction force $F_{0}=F_{1p}+F_{1a}$, which can be seen in Figure 9B.


*Start walking:*


In the beginning of walking, with respect to the muscles, for flexion of the hip the rectus femoris contracts and for dorsiflexion the tibialis anterior contracts. With respect to the ground reaction force, the initial value $F_{0}$ increases in the part on the supporting foot (in this case on the right foot) $F_{d}$ and the force on the left foot $F_{i}$ decreases, where $F_{d}>F_{i}$ and $F_{d}>F_{0}$, which can be seen in Figure 9C.

#### Signal Threshold Detection

The signal threshold detection method uses the physiological description of sEMG signals from the muscles and the physics of FGR sensors found in Section 4.1.

When a person gets up from a chair, the gastrocnemius contracts to perform plantar flexion which is seen in Figure 10. The sEMG for gastrocnemius (line in color pink) starts to increment its amplitude indicating the intention of motion to get up. Then, rectus femoris muscle that is an extensor of the knee contracts indicating the end of the motion. With respect to the ground reaction force in Figure 11, the FGR signals at the front part of both feet, left front (LF), and right front (RF) start to increase. It indicates the initial force when a person uses the front part of feet to get up.

The identification percentages of the three movement routines are presented in Table 3. Furthermore, the confusion matrix to evaluate the performance of the detection is shown in Table 4.

## 5. Feedback Control

In this section, the control approach is described. Let us remember that: first, the human intention is detected using electromyograpy and baropodometry, then the system chooses which task will be developed (sitting to standing, standing to sitting, or sitting to walk), and selects one of the desired trajectories $q_{\mathrm{dSTS}}$, $q_{\mathrm{dSTA}}$ or $q_{\mathrm{dSW}}$, respectively, which were obtained by video analysis and are already programmed in the exoskeleton’s computer.

To obtain the control law, the system is considered to be connected in a cascade of the dynamics of the robot links and the dynamics of the motors links. The dynamics of the links is actuated by the angles of the motors $q_{m}$ through the flexible joints, the dynamics of the motors is actuated by the motor torques τ. From Equation (Equation 5), which corresponds to the dynamical model, we have:(13)$$\begin{array}{cc} \; & M\left(q_{e}\right){\ddot{q}}_{e}+C\left(q_{e},{\dot{q}}_{e}\right){\dot{q}}_{e}+g\left(q_{e}\right)+Kq_{e}=Kq_{m}\\ & {\ddot{q}}_{m}=J^{-1}\left(\tau -K\left(q_{m}-q_{e}\right)-B{\dot{q}}_{m}\right). \end{array}$$

The methodology used [24] considered $q_{m}$ as the input of the first equation in (Equation 13) and a control law $q_{m}^{\mathrm{des}}$ ∈ $R^{2}$ is proposed for $q_{m}$ as:(14)$$q_{m}^{\mathrm{des}}=q_{e}+K^{-1}\left[M\left(q_{e}\right)\dot{v}+C\left(q_{e},{\dot{q}}_{e}\right)v+g\left(q_{e}\right)-K_{d}r\right],$$
where $K_{d}>0$ ∈ $R^{2\times 2}$ is diagonal and constant,



$$v={\dot{q}}_{e}^{\mathrm{des}}-\lambda_{1}{\tilde{q}}_{e},$$





$${\tilde{q}}_{e}=q_{e}-q_{e}^{\mathrm{des}},$$



$$r={\dot{q}}_{e}-v$$
where v, ${\tilde{q}}_{e}$ and r ∈ $R^{2}$, $\lambda_{1}>0$ ∈ $R^{2\times 2}$ is diagonal and constant.

Defining ${\tilde{q}}_{m}=q_{m}-q_{m}^{\mathrm{des}}$ ∈ $R^{2}$, and substituting in the first equation of (Equation 13) we have:(15)$$M\left(q_{e}\right)\dot{r}+C\left(q_{e},{\dot{q}}_{e}\right)r+K_{d}r=K{\tilde{q}}_{m}.$$

Lyapunov’s candidate function is chosen as:(16)$$V_{1}=\frac{1}{2}r^{T}M\left(q_{e}\right)r.$$

The derivative along the trajectories of the system is:(17)$${\dot{V}}_{1}=\frac{1}{2}r^{T}\dot{M}\left(q_{e}\right)r+r^{T}M\left(q_{e}\right)\dot{r}=-r^{T}K_{d}r+r^{T}K{\tilde{q}}_{m}.$$

If ${\tilde{q}}_{m}=0$, then ${\dot{V}}_{1}<0$ and r tend to zero when t→0 and as $r={\dot{\tilde{q}}}_{e}+\lambda_{1}{\tilde{q}}_{e}$ with, so ${\dot{\tilde{q}}}_{e}$ and ${\tilde{q}}_{e}$ have zero when t→∞.

As a second step, $q_{m}$ is considered as the input of the second equation of (Equation 13). Derive ${\tilde{q}}_{m}$ as ${\dot{\tilde{q}}}_{m}={\dot{q}}_{m}-{\dot{q}}_{m}^{\mathrm{des}}$ and ${\ddot{\tilde{q}}}_{m}={\ddot{q}}_{m}-{\ddot{q}}_{m}^{\mathrm{des}}$, substituting the above into (Equation 13) we have:(18)$${\ddot{\tilde{q}}}_{m}=J^{-1}\left(\tau -B{\dot{q}}_{m}+K\left(q_{e}-q_{m}\right)\right)-{\ddot{q}}_{m}^{\mathrm{des}}.$$

The proposed control law is:(19)$$\tau =-JK_{1}{\dot{\tilde{q}}}_{m}-JK_{2}{\tilde{q}}_{m}+Jq_{m}+B{\dot{q}}_{m}-K\left(q_{e}-q_{m}\right)),$$
where $K_{1}$ and $K_{2}$ are a positive diagonal matrix.

Then,
(20)$$J^{-1}\left(\tau -B{\dot{q}}_{m}+K\left(q_{e}-q_{m}\right)\right)-{\ddot{q}}_{m}=-K_{1}{\dot{\tilde{q}}}_{m}-K_{2}{\tilde{q}}_{m};$$
therefore,
(21)$${\ddot{\tilde{q}}}_{m}=-JK_{1}{\dot{\tilde{q}}}_{m}-JK_{2}{\tilde{q}}_{m}.$$

Lyapunov’s candidate function is chosen as:(22)$$V_{2}=\frac{1}{2}J^{-1}{\dot{\tilde{q}}}_{m}^{2}+\frac{1}{2}{\tilde{q}}_{m}^{2}.$$

The derivative along the trajectories of the system is:(23)$$\begin{array}{cc} {\dot{V}}_{2} & =J^{-1}{\ddot{\tilde{q}}}_{m}{\dot{\tilde{q}}}_{m}+{\dot{\tilde{q}}}_{m}{\tilde{q}}_{m}\\ \; & =-{\dot{\tilde{q}}}_{m}K_{1}{\dot{\tilde{q}}}_{m}-{\dot{\tilde{q}}}_{m}\left(K_{2}-I\right){\dot{\tilde{q}}}_{m}, \end{array}$$
where *I* is the identity matrix.

If $K_{2}>I\Rightarrow {\dot{V}}_{2}<0$, then ${\tilde{q}}_{m}\to 0,{\dot{\tilde{q}}}_{m}\to 0,{\ddot{\tilde{q}}}_{m}\to 0$, from (Equation 16), it follows r→0 then ${\tilde{q}}_{e}\to 0$.

## 6. Results

### 6.1. Numerical Results

#### 6.1.1. Detection Algorithm Implementation

The detection algorithm for the human intention was performed and tested offline with the signals obtained from the sEMG and FGR sensors of the system described in Section 4.1. The database contains 10 s long signals obtained from a healthy female subject during the performance of the three movements: sitting to standing, standing to sitting and standing to walking, which were acquired with a sampling frequency of 1 kHz along 10 trials. The 10 trials are enough to prove the detection algorithm due to the behavior in four different recording sessions, which were performed in the same way. The total number of tasks considering the three movements is equal to 30, which generate muscular fatigue in the patient. Due to the muscular fatigue, we recorded the 30 trials per session because more trials could degrade the signal behavior. In Figure 10, a record of sEMG normalized signals from sitting to standing is shown. The four channels correspond to the four muscles: Rectus femoris (RF) in red, Biceps Femoral (BF) in blue, Tibial anterior (TA) in green and Gastrocnemius-Gas in pink. The envelopes of electromyographic signals were obtained by the MyoWare^®^ sensors and the increment of amplitude represents the contraction of the muscle. The FGR normalized signals from sitting to standing of four sensors corresponding to the position Left Front (LF), Left Back (LB), Right Front (RF), and Right Back (RB) are shown in Figure 11. The increment of FGR signals represents the force exerted by both feet.

#### 6.1.2. Implementation of the Exoskeleton’s Control

The simulation of the control applied to the lower limb robot was carried out in Matlab-simulink. The desired position was entered with a column vector with the values of the two desired joints $q_{d1}$ and $q_{d2}$. The control law of Equations (Equation 14) and (Equation 19) was simulated together with the dynamical model of the Equation (Equation 5) considering the parameters indicated in Table 5.

The initial conditions of the angular positions are chosen as: $q_{1}\left(0\right)=0$ and $q_{2}\left(0\right)=0$ and the control gains used are: $\lambda_{1}=diag\left\{10,10\right\}$ and Kd=diag{1,1}.

The results of the trajectory tracking are shown in Figure 12 and Figure 13. The desired trajectories are shown in blue, the trajectory of the links is shown in black and the trajectory of the motors in red. The tracking position is displayed for the Sitting to Standing (STS), Standing to Sitting (STA), and Standing to Walking (SW) movement routines.

The trajectories of the links $q_{1}$ and $q_{2}$ (line in color black) tend to the desired trajectories $q_{1d}$ and $q_{2d}$ (dotted line in color blue). The trajectories of the motors $q_{1m}$ and $q_{2m}$ (dotted line in color red) tend to the desired trajectory with noise, this behavior is due to the springs considered in the elastic joints (Figure 12 and Figure 13).

For the hip joint, the variation of angular position is greater compared to the knee joint due to the inertia to be moved.

The results of the tracking errors denoted as $e_{q}=q_{d}-q$ and $e_{\mathrm{qm}}=q_{d}-q_{m}$ corresponding to the joins motion with and without the model using elastic joins respectively are shown in Figure 14 and Figure 15.

The tracking error of the $e_{q}$ links motion (line in color black) oscillate around zero without a lot of noise; the tracking error of the motors $e_{\mathrm{qm}}$ (line in color blue) varies due to spring behavior.

### 6.2. Real-Time Experimental Results

This section presents the results obtained from the experimental evaluation for tracking control implemented on a National Instruments myRIO data acquisition card, and programmed in LabView software.

The human intention identification system was proved with a healthy female subject of 27 years with a weight of 55 kg and a height of 1.6 m.

The objective of the controller is the tracking trajectory and the flexible joint exoskeleton performs the movement corresponding to the identified task, see Figure 16.

The exoskeleton trajectory tracking is shown at the top of Figure 17. The Stand to Walking movement (SW) achieved by the hip is shown in Figure 17A and by the knee in Figure 17B. The join trajectories for Stand to Walking are denoted by $q_{1}SW$ and $q_{2}SW$ (dotted line in blue), and the join desired trajectories are denoted by $q_{1d}SW$ and $q_{2d}SW$ (line in black). The trajectories of the motor are denoted by $q_{1m}SW$ and $q_{2m}SW$.

Subsequently, Figure 17C,D show the exoskeleton movement from Sit to Stand (STS) carried out by joins $q_{1}STS$ and $q_{2}STS$ (dotted line in blue) that follow the desired trajectories $q_{1d}STS$ and $q_{2d}STS$ (line in black) likewise.

## 7. Conclusions

A motion intention detection system was obtained using electromyography sensors and ground reaction force sensors that operate as a baropodometry sensor. Four muscles were established whose signals are read by the detection system: the Biceps Femoris (BF), the Rectus Femoris (RF), the Tibialis Anterior (TA) and the Gastrocnemius (GAS).

The human intention detection system has a performance of 90% of correct detection. In addition, the performance metrics show an average precision of 0.9, indicating a good performance of the system.

An important contribution of this paper is to show an effective way to obtain the human intention using electromyography and baropodometry signals. However, sensing the intention of movement mechanically is essential for other types of active exercises. For this reason the exoskeleton was designed with this type of flexible actuators containing a torque sensor.

The required trajectories for motion in an exoskeleton’s joints were obtained from the analysis of movement of different videos, which were approximated by trigonometric polynomials. The trajectories in the implementation of the exoskeleton control were suitable for 3D-CAD motion.

The proposed control law is based on the exoskeleton dynamic model; this exoskeleton mobilizes the hip and knee joints. The implemented control has a good performance for tracking trajectory, which means that tracking errors tend to zero. In fact, when a rehabilitation is performed, the most convenient is to adjust the control gains, this action is very simple, only tuning the gains in the sense of closed-loop stability ($K_{1}$ must be positive and $K_{2}>I$). The error in the link model is smaller than in the motor model, which is explained by the dynamics of the elastic joints in motors.

## Figures and Tables

**Figure 1 sensors-23-05252-f001:**
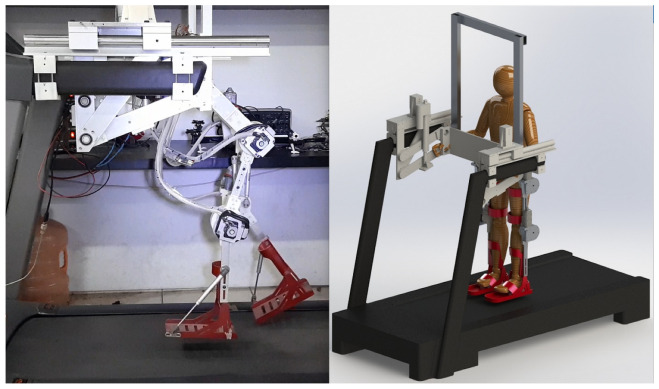
On the left is the real prototype of the exoskeleton for gait rehabilitation during a walking routine and on the right is the CAD design of the prototype with a full view.

**Figure 2 sensors-23-05252-f002:**
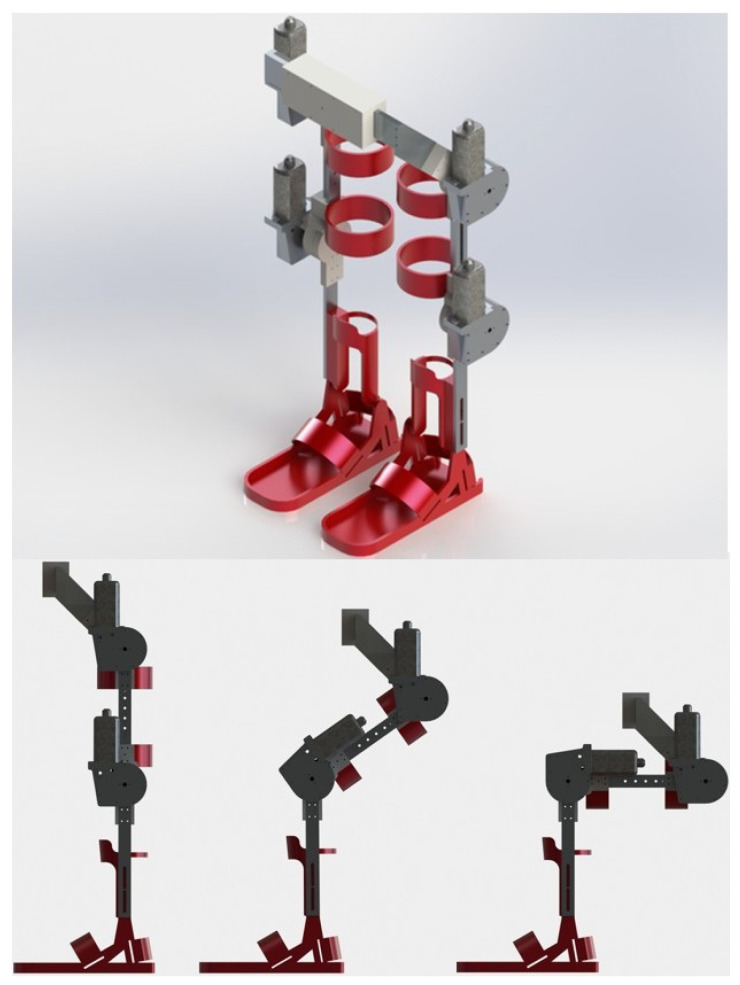
CAD drawing of the exoskeleton legs. In the upper part there is an isometric view and in the lower part a side view of the sitting process.

**Figure 3 sensors-23-05252-f003:**
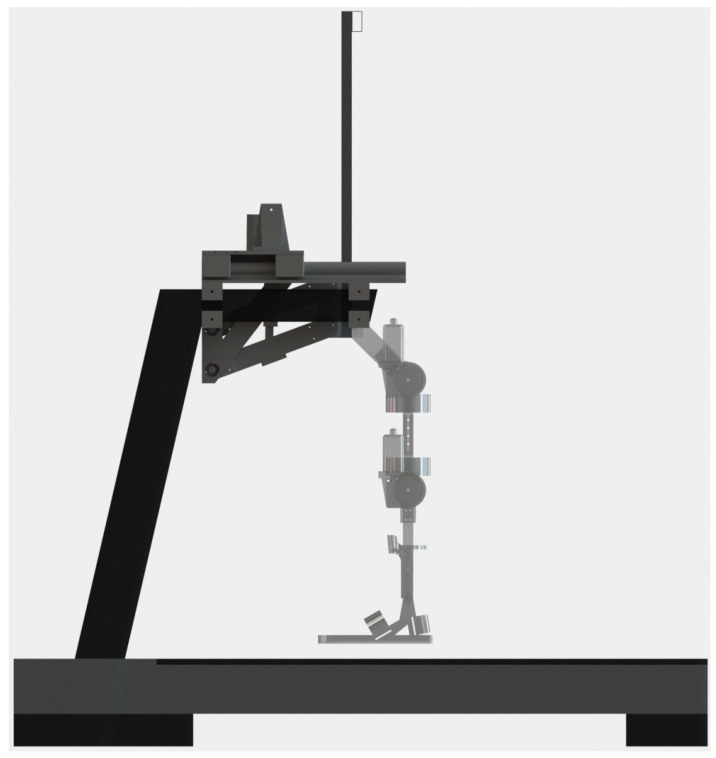
CAD drawing of the exoskeleton lifting system consisting of a double four-bar system operated by two linear actuators.

**Figure 4 sensors-23-05252-f004:**
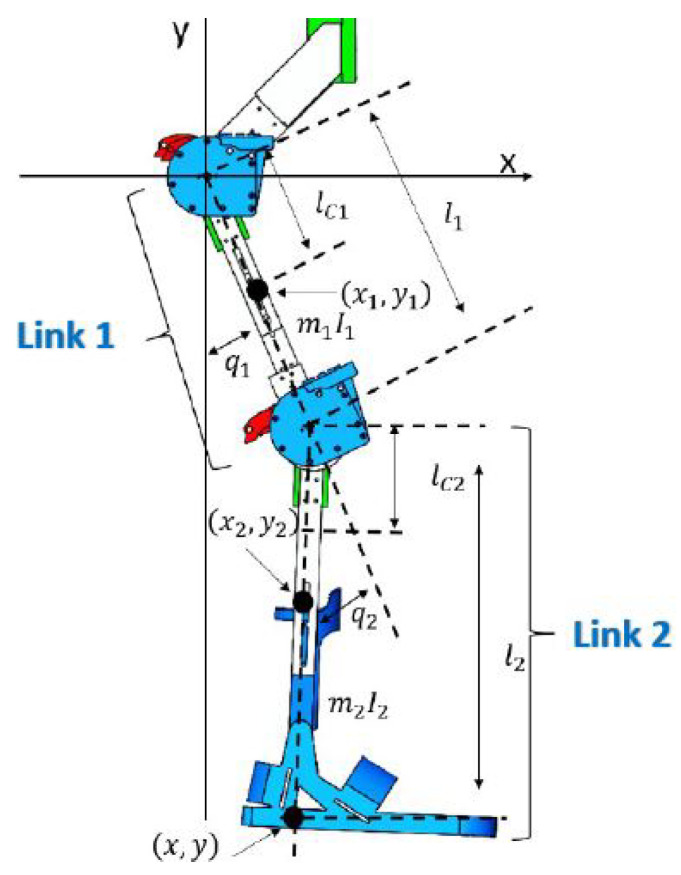
Schematic diagram of the exoskeleton; link 1 corresponds to the thigh and link 2 corresponds to the leg with lengths $l_{1}$ and $l_{2}$, and with masses $m_{1}$ and $m_{2}$, respectively.

**Figure 5 sensors-23-05252-f005:**
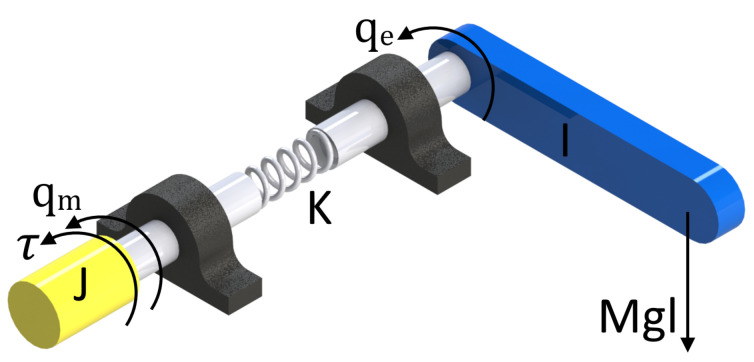
AD schematic of an elastic rotational actuator, $q_{m}$ is the angular position of the motor, and $q_{e}$ is the angular position of the link.

**Figure 6 sensors-23-05252-f006:**
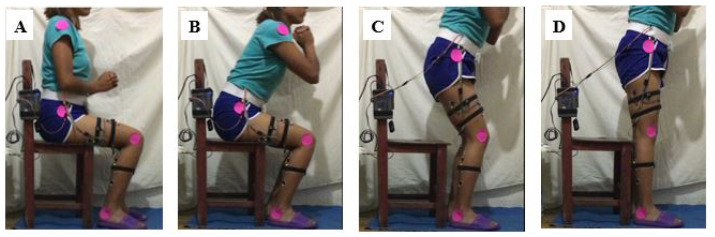
Frames of video analysis to obtain the trajectories of the cycle from sitting to standing, using markers in hip, knees, and ankles; (**A**) Fully seated, (**B**) Lean to stand, (**C**) Bow on Hip, knees and ankles, and (**D**) Fully standing.

**Figure 7 sensors-23-05252-f007:**
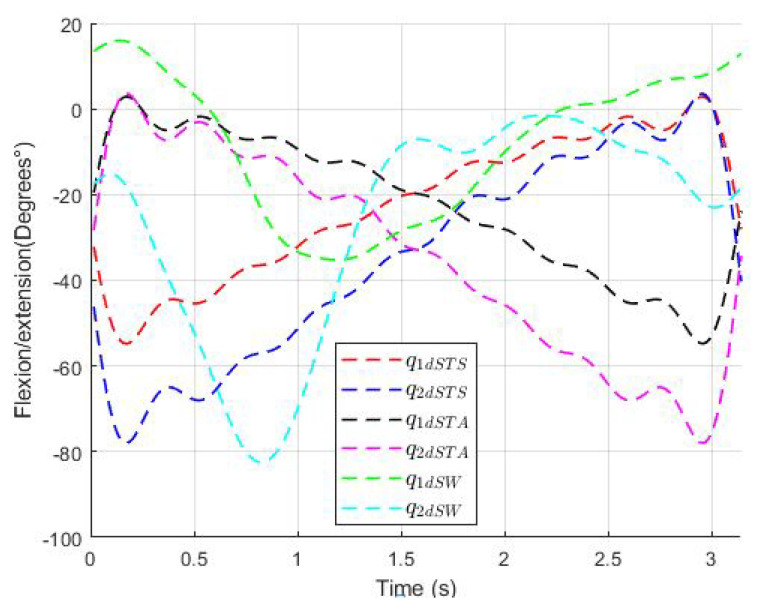
Motion trajectories for exoskeleton, sitting to standing $q_{d}\mathrm{STS}$, standing to sitting $q_{d}\mathrm{STA}$, and standing to walk $q_{d}\mathrm{SW}$, to the hip and knee joint denoted by $q_{d1}$ and $q_{d2}$, respectively.

**Figure 8 sensors-23-05252-f008:**
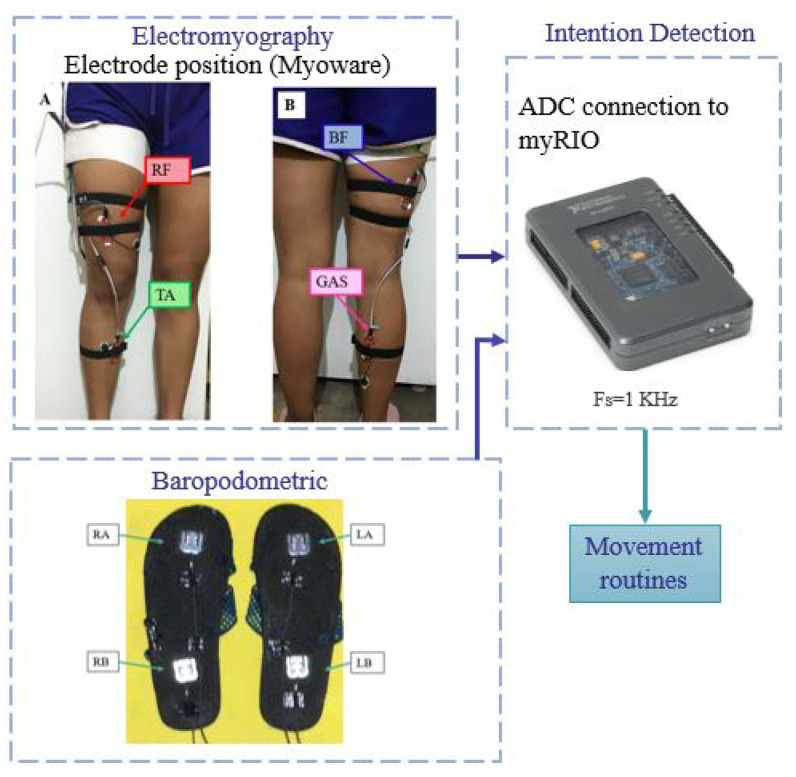
Block diagram representing the human intention identification system, integrated of electromyography sensors (A. Front view and B. Rear view), baropodometry sensors and the data acquisition card that sends the signals to the classifier in the computer.

**Figure 9 sensors-23-05252-f009:**
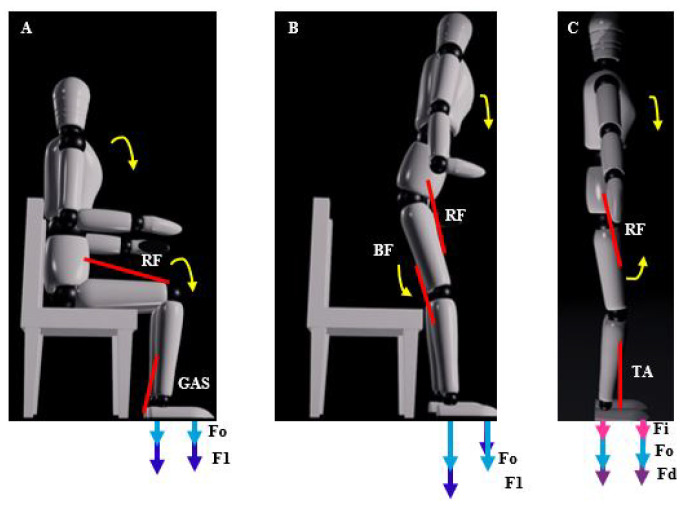
Effects obtained from the human intention detection system for: (**A**) Getting up, (**B**) Sitting, and (**C**) Starting to walk.

**Figure 10 sensors-23-05252-f010:**
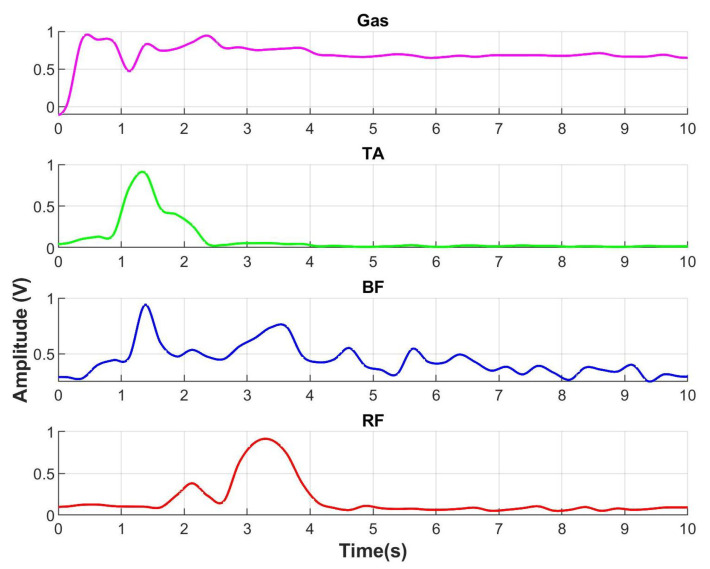
Electromyographic signals obtained from the movement from sitting to standing, Rectus femoris (RF) in red, Biceps Femoral (BF) in color blue, Tibial anterior (TA) in green, and Gastrocnemius (Gas) in pink.

**Figure 11 sensors-23-05252-f011:**
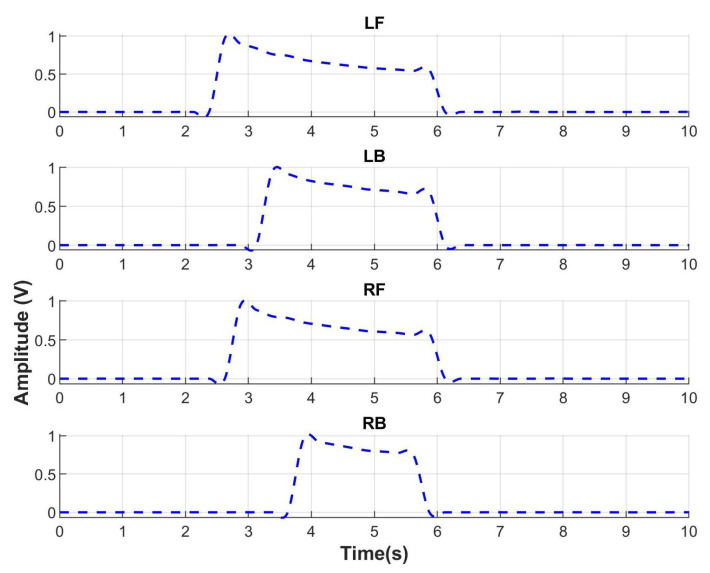
FGR signals of the movement from sitting to standing, Left Front (LF), Left Back (LB), Right Front (RF), and Right Back (RB).

**Figure 12 sensors-23-05252-f012:**
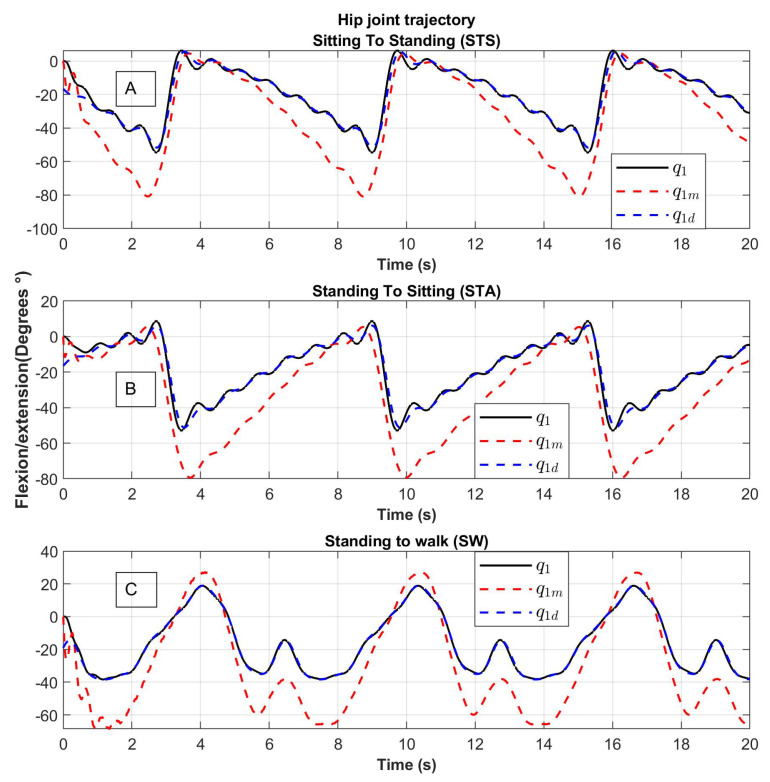
Hip joint tracking of movement from (**A**) Sitting to Standing (STS), (**B**) Standing to Sitting (STA) and (**C**) Standing to Walking (SW); $q_{1}$: the trajectory of the link, $q_{1d}$: desired trajectory, $q_{1m}$: the trajectory of the motor.

**Figure 13 sensors-23-05252-f013:**
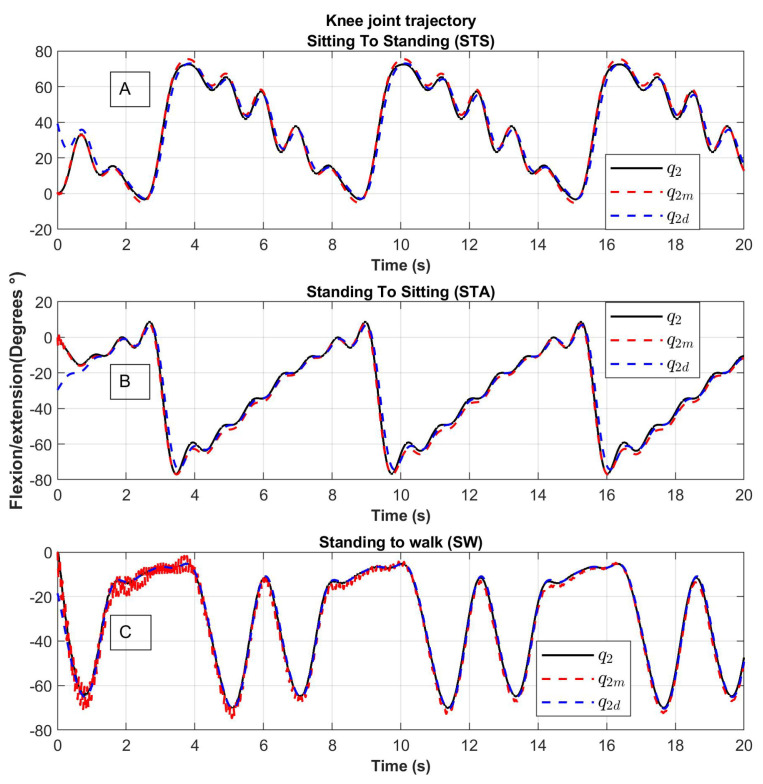
Knee joint tracking of movement from (**A**) Sitting to Standing (STS), (**B**) Standing to Sitting (STA) and (**C**) Standing to Walking (SW); $q_{1}$: the trajectory of the link, $q_{1d}$: desired trajectory, $q_{1m}$: the trajectory of the motor.

**Figure 14 sensors-23-05252-f014:**
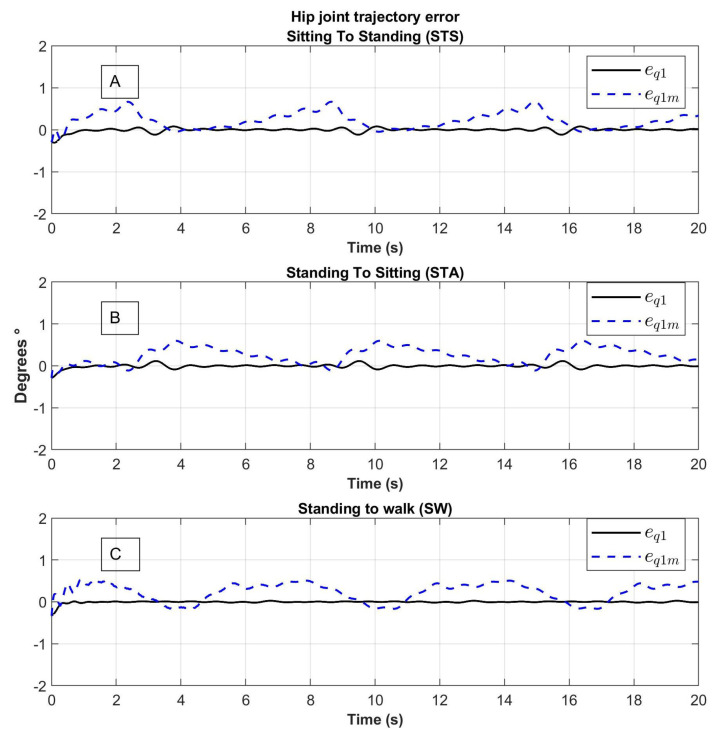
Hip joint tracking error of movement from (**A**) Sitting to Standing (STS), (**B**) Standing to Sitting (STA) and (**C**) Standing to Walking (SW); $e_{q1}$: error of the trajectory of the link, $q_{1m}$: error of the trajectory of the motor.

**Figure 15 sensors-23-05252-f015:**
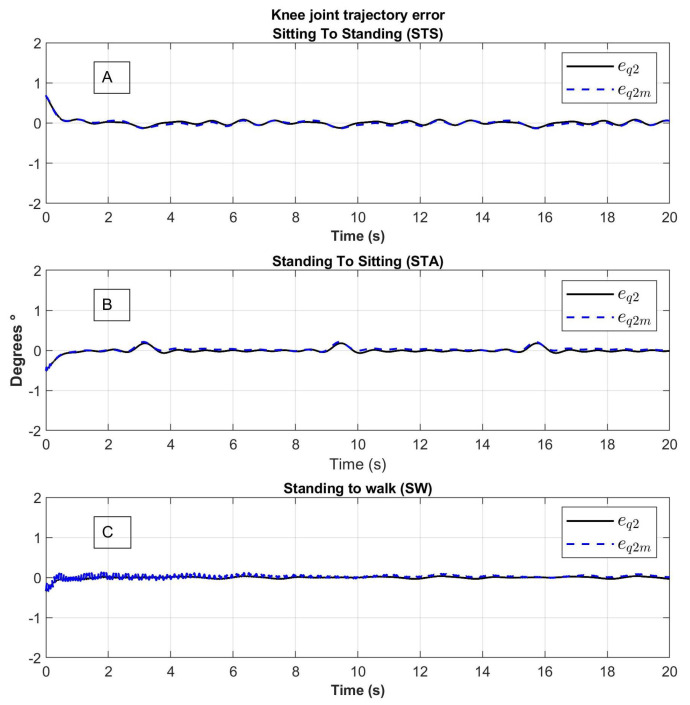
Knee joint tracking error of movement from (**A**) Sitting to Standing (STS), (**B**) Standing to Sitting (STA) and (**C**) Standing to Walking (SW); $e_{q1}$: error of the trajectory of the link, $q_{1m}$: error of the trajectory of the motor.

**Figure 16 sensors-23-05252-f016:**
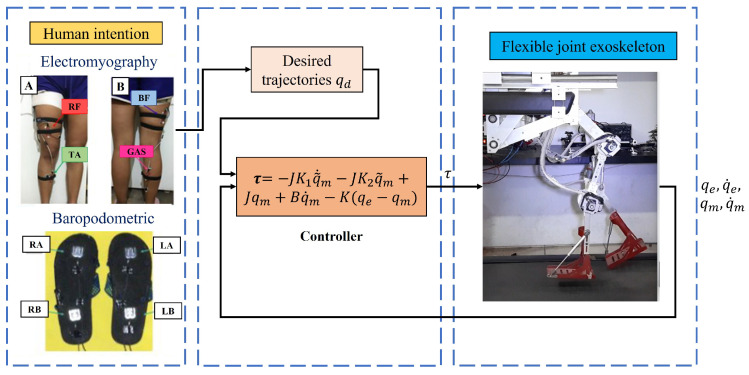
Block diagram that represents the control system, integrated by the human detection system, integrated by electromyography (A. Front view and B. Rear view) and baropodometry, the control law and the exoskeleton prototype.

**Figure 17 sensors-23-05252-f017:**
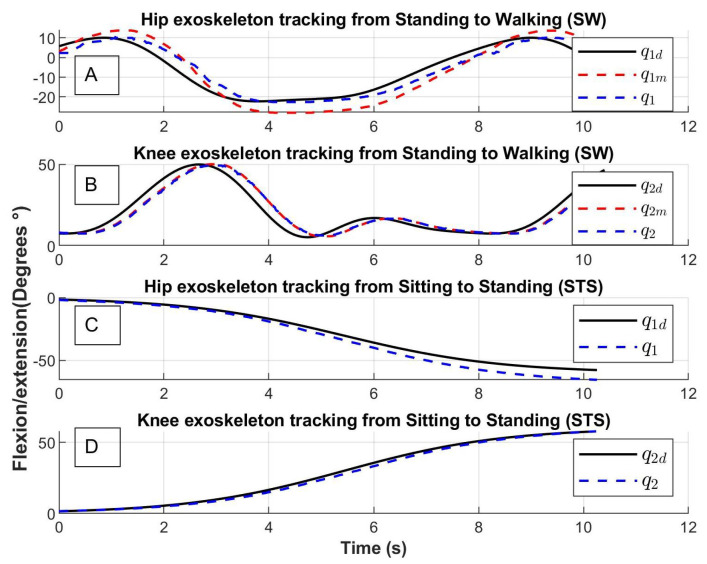
Exoskeleton tracking trajectory of (**A**) Hip from Standing to Walking (SW), (**B**) Knee from Standing to Walking (SW), (**C**) Hip from Sitting to Standing (STS) and (**D**) Knee from Sitting to Standing (STS); $q_{i}$: the trajectory of the link, $q_{id}$: desired trajectory, $q_{im}$: the trajectory of the motor with i=1,2.

**Table 1 sensors-23-05252-t001:** Supported range of motion for the leg joints.

	RoM for the Human [19]		RoM for the Robot	
**Joint**	**Minimum Angle**	**Maximum Angle**	**Minimum Angle**	**Maximum Angle**
Hip	$-22^{\circ }$	$122^{\circ }$	$-14^{\circ }$	$90^{\circ }$
Knee	$0^{\circ }$	$134^{\circ }$	$0^{\circ }$	$110^{\circ }$

**Table 2 sensors-23-05252-t002:** Upper bound in trajectories.

Routine	Trajectories	Upper Bound
Sitting to standing	$q_{d1STS}\left(t\right)$	51.6276 deg
	$q_{d2STS}\left(t\right)$	72.9801 deg
	${\dot{q}}_{d1STS}\left(t\right)$	118.5088 deg/s
	${\dot{q}}_{d2STS}\left(t\right)$	115.8713 deg/s
Standing to sitting	$q_{d1STA}\left(t\right)$	51.1969 deg
	$q_{d2STA}\left(t\right)$	74.2387 deg
	${\dot{q}}_{d1STA}\left(t\right)$	118.7486 deg/s
	${\dot{q}}_{d2STA}\left(t\right)$	167.3876 deg/s
Standing to walking	$q_{d1SW}\left(t\right)$	38.3282 deg
	$q_{d2SW}\left(t\right)$	70.3928 deg
	${\dot{q}}_{d1SW}\left(t\right)$	66.8698 deg/s
	${\dot{q}}_{d2SW}\left(t\right)$	99.3652 deg/s

**Table 3 sensors-23-05252-t003:** Identification percentages.

Identification Percentages	STS	STA	SW
% correct	90%	80%	100%
% incorrect	10%	20%	0%

**Table 4 sensors-23-05252-t004:** Confusion matrix.

	STS Actual	STA Actual	SW Actual	Total
STS Predicted	9	2	0	11
STA Predicted	1	8	0	9
SW Predicted	0	0	10	10
Total	10	10	10	

**Table 5 sensors-23-05252-t005:** Simulation parameters.

Variable	Meaning	Value
$m_{1}$	Link mass 1	0.5 kg
$m_{2}$	Link mass 1	0.6 kg
$I_{1}$	Link moment of inertia 1	0.12 kg m${}^{2}$
$I_{2}$	Link moment of inertia 2	0.12 kg m${}^{2}$
$l_{1}$	Link length 1	0.26 m
$l_{2}$	Link length 2	0.30 m
$l_{c1}$	Link center of mass length 1	0.026 m
$l_{c2}$	Link center of mass length 2	0.026 m

## Data Availability

https://umi.cinvestav.mx/Equipos-de-Investigacion/Exoesqueletos.

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
