# Peer review of "Lower Limb Exoskeleton for Rehabilitation with Flexible Joints and Movement Routines Commanded by Electromyography and Baropodometry Sensors"

_sensors, 2023, doi:10.3390/s23115252_

Round 1
Reviewer 1 Report
Minor revision
ž The flow of paragraphs in the introduction section is not smooth, so it does not lead to a point.
ž Detection algorithm and classification algorithm look the same, but want to be unified.
(7.1, The motion intension detection algorithm is performed and tested offline-
7.2 - The confusion matrix to validate the performance of the classification algorithm is show in table 3)
ž There is a big difference in the overall literary style.
ž I would like an explanation of the overall figure legend to be added under figure in addition to the body. (ex. Figure 12. Hip joint tracking of movement Hip joint tracking of movement from A) Sitting to Standing (STS), B) Standing to Sitting (STA) and C) Standing to Walking (SW) q1: the trajectory of the links, q1d: desired trajectory, q1m: the trajectory of the motors)
ž Table 1 displays the Range of motion that people need, but for comparison, I would like you to add RoM for the general person and RoM for the robot. (ex. Human maximum & minimum joint angle, Allowed angle, Exoskeleton limit)
Major revision
ž Tables 2 and 3 show the accuracy of behavior recognition and the results of experiments. The experiment was conducted 10 times for each action, and the results were reflected in accuracy, but 10 times for each of the three actions is not enough. In addition, information about the experimenter is not reflected.
ž In explaining the "control system for understanding the wearer's intentions," there is a lack of explanation for the classification control technique. I would also like to add a control block diagram (I think you can put it in before ‘7.3 Implementation of the exoskeleton's control’ comes out).
ž If you look at the experimental results of Figure 14's hip joint tracking error, it is unlikely that the tracking error will converge to zero.
ž In verifying the Contributions of this paper, it is thought that the overall experiment type is insufficient. I also feel that there is a lack of analysis of the results of the experiment and the conclusion of the whole paper.
Author Response
We appreciate that you gave us the opportunity to continue in the process of publishing this paper, we appreciate your time for your valuable review and above all we appreciate all your comments, and we will attend each of your observations.
You will find attached the document of the revised paper with all the changes highlighted in yellow.
Comments and Suggestions for Authors:
Reviewer 1
Minor revision
Reviewer 1: The flow of paragraphs in the introduction section is not smooth, so it does not lead to a point.
Answer: The Introduction section was improved in the sense to highlight our objectives.
Reviewer 1: Detection algorithm and classification algorithm look the same, but want to be unified.
(7.1, The motion intension detection algorithm is performed and tested offline-
7.2 - The confusion matrix to validate the performance of the classification algorithm is show in table 3)
Answer: The terms detection and classification were unified in the paper, remaining only as: detection algorithm.
Reviewer 1: There is a big difference in the overall literary style.
Answer: The complete paper was revised to improve the literary style, including bibliography and references.
Reviewer 1: I would like an explanation of the overall figure legend to be added under figure in addition to the body. (ex. Figure 12. Hip joint tracking of movement Hip joint tracking of movement from A) Sitting to Standing (STS), B) Standing to Sitting (STA) and C) Standing to Walking (SW) q1: the trajectory of the links, q1d: desired trajectory, q1m: the trajectory of the motors)
Answer: To give a better explanation of the figures, information in the legends of the figures was added: 3 through 17.
Reviewer 1: Table 1 displays the Range of motion that people need, but for comparison, I would like you to add RoM for the general person and RoM for the robot. (ex. Human maximum & minimum joint angle, Allowed angle, Exoskeleton limit)
Answer: In table 1 we added the ranges of movement RoM of the human.
Major revision
Reviewer 1: Tables 2 and 3 show the accuracy of behavior recognition and the results of experiments. The experiment was conducted 10 times for each action, and the results were reflected in accuracy, but 10 times for each of the three actions is not enough. In addition, information about the experimenter is not reflected.
Answer: To clarify, the experiments were performed only 10 times for each action (STS, STA and SW) presented in Tables 2 and 3. The following explanation was added (line 368-376):
“The 10 trials are enough to prove the detection algorithm due to the behavior in 4 different recording session which were performed in the same way. The total number of tasks considering the three movements is equal to 30, which generate muscular fatigue in the patient. Due to the muscular fatigue, we recorded the 30 trials per session because more trials could degrade the signal behavior.”
Reviewer 1: In explaining the "control system for understanding the wearer's intentions," there is a lack of explanation for the classification control technique. I would also like to add a control block diagram (I think you can put it in before ‘7.3 Implementation of the exoskeleton's control’ comes out).
Answer: We added Figure 16 in section 7.3 in order to explain the classification control technique.
In addition, to better explain the classification control technique, the following paragraph was added (line 328-332):
“In this section the control approach is described. Let us remember that: first, the human intention is detected using electromyography and baropodometry, then the system chooses which task will be developed (Sitting to standing, Standing to sitting or Sitting to walk), and selects one of the desired trajectories qdSTS, qdSTA or qdSW respectively, which were obtained by video analysis and are already programmed in the exoskeleton's computer.”
Reviewer 1: If you look at the experimental results of Figure 14's hip joint tracking error, it is unlikely that the tracking error will converge to zero.
Answer: In the simulation results section, the graphs of Figures 14 and 15 were modified to show better results with respect to the convergence to zero of the hip and knee joints tracking error.
Reviewer 1: In verifying the Contributions of this paper, it is thought that the overall experiment type is insufficient. I also feel that there is a lack of analysis of the results of the experiment and the conclusion of the whole paper.
Answer: In the experiments section, the graphs of Figure 17 and their corresponding analysis were modified to show better results.

Reviewer 2 Report
While I find the research topic interesting, there are some aspects of it that are not appropriate for a manuscript.
#1
The method of EMG analysis needs significant revision. The following statements are not identified in the manuscript.
- You describe the required measurement frequency from the Nyquist frequency, but I do not know the actual measurement frequency used in the EMG.
- Please describe the filter applied to the EMG data
- The smoothed EMG data contains noise or appears to drift (especially the Gastrocnemius EMG is highly inappropriate as data for publication).
- Smoothing process for EMG
#2
Please try to be consistent about the detailed formatting.
In particular, we see some figures that do not fit in the frame of the Table. Is there an intention to do so?
#3
There is no caption on the y-axis of Figure10,11.
Author Response
We appreciate that you gave us the opportunity to continue in the process of publishing this paper, we appreciate your time for your valuable review and above all we appreciate all your comments, and we will attend each of your observations.
You will find attached the document of the revised paper with all the changes highlighted in yellow.
Comments and Suggestions for Authors:
Reviewer 2: The method of EMG analysis needs significant revision. The following statements are not identified in the manuscript.
You describe the required measurement frequency from the Nyquist frequency, but I do not know the actual measurement frequency used in the EMG.
Answer: The measurement frequency was added in paragraph: line 278 “Then, the used sampling frequency is of 1 kHz.”
Reviewer 2: Please describe the filter applied to the EMG data
Answer: A description of the filter was added in the paragraph: line 278-279 “The signals are filtered using a band-pass filter type butter-worth of fourth order from 10 to 500 Hz to have better information.”
Reviewer 2: The smoothed EMG data contains noise or appears to drift (especially the Gastrocnemius EMG is highly inappropriate as data for publication). Smoothing process for EMG.
Answer: We improved the graph of Figure 10 to show better results with respect to the noise in the EMG signals.
Reviewer 2: Please try to be consistent about the detailed formatting.
Answer: The complete paper was revised to improve the formatting, including bibliography and references.
Reviewer 2: In particular, we see some figures that do not fit in the frame of the Table. Is there an intention to do so?
Answer: All the Tables were corrected
Reviewer 2: There is no caption on the y-axis of Figure 10,11.
Answer: Figures 10, 11 and the other Figures were revised to correctly add legends of the y-axis graphs.

Reviewer 3 Report
First, I will focus on the general aspects of the paper. It is an interesting work that combines theoretical and practical aspects to detail an exoskeleton specialized in lower limb rehabilitation. The paper includes mechanical design, sensorics, trajectory generation and control strategy. In this sense, it is a very complete work. However, there are certain aspects that need to be improved. Specifically, the main weaknesses of the article are: i) It does not justify why it does not affect the dynamics of the human body for the control strategy design; ii) It does not detail how to estimate, with an algorithm, the intention of movement and how this becomes the control reference; iii) It does not study the trajectories properties for following a dynamics of both speed and acceleration and iv) It does not present any control specification for the adjustment of the controller. Now, I detail the issues section by section.
1. Introduction
It mixes the state of the art with the description of the exoskeleton they have developed. This section should be devoted only to the state of the art and highlight the new contributions of the work. Also, figure 1 does not present the same exoskeleton configuration as described later. It is confusing.
2. Mechanical construction.
This section should include a complete description of the developed exoskeleton. It is also convenient to include a control architecture and its description to link all the elements that will be described in the following sections.
3. Mathematical model.
The first question to be answered is why the dynamics of the person who will use the exoskeleton is not taken into account, and only the dynamics of the exoskeleton is taken into account. By methodology, before describing the dynamic model, it is convenient to present the direct kinematic model and the Jacobian model before approaching the calculation of the dynamic model. The construction of the dynamic model considers the kinetic and potential energy of the mechanism, making the calculations in formulae (3) and (4) redundant. It is sufficient to include the actuator with the elastic element in the dynamic model. Furthermore, the calculations do not include the gearbox, the effect of which is to make the system linear and thus convert it into a linear MIMO system. Thus, the stability study and the design of the controller are much simpler. Everything described in subsection 3.1.1. is redundant with respect to the calculations performed to build the dynamic model of the mechanism. With respect to the generation of the trajectories, the dynamic model is not used to constrain them in velocity and acceleration. It is important to take this into account in order not to saturate the actuators or to establish a comfortable dynamic of movement for the patient. The calculation of the trajectory generation is disconnected from the dynamic model. Another important issue is that the dynamic model is not validated. Are the parameters of the dynamic model correct? Has an estimation procedure been performed to calculate the parameters of the dynamic model?
4. Sensors.
This section should be linked to section 5, dedicated to intention detection. It should also include the specific algorithm used to convert this intention of movements into commands for trajectory generation or control.
5. Human intention.
This requires further formalization and linking to the previous section. It does not detail anything about the filtering of the signals or the classification algorithm to be used to detect the intention. A sensory system architecture is also needed.
6. Feedback control.
The authors are strongly based on a control strategy and stability analysis already published in "Modeling and control of robot manipulators" by L. Sciavicco and B. Siciliano, ISBN 0-07-057217-8. Siciliano, ISBN 0-07-057217-8. So, this section can be summarized and refer to this book. The point is that the dynamic model, when converted into a MIMO system, can simplify the stability analysis. Other unanswered questions: What are the control specifications used to calculate the controller gains?, How do the dynamics of the human body affect the controller performance), Is it necessary to consider a shared control scheme?.
7. Numerical results.
This section and section 8 can be merged. How the signals are used to plan the exoskeleton trajectories?. In figure 10, the units of the "Y" axis are missing. Section 7.2. should be part of section 5.
8. Real-time experimental results.
The conditions under which each experiment is performed should be specified. What is the weight, height of the patient using it? Does this affect the performance of the controller?
Author Response
We appreciate that you gave us the opportunity to continue in the process of publishing this paper, we appreciate your time for your valuable review and above all we appreciate all your comments, and we will attend each of your observations.
You will find attached the document of the revised paper with all the changes highlighted in yellow.
Comments and Suggestions for Authors:
Reviewer 3: First, I will focus on the general aspects of the paper. It is an interesting work that combines theoretical and practical aspects to detail an exoskeleton specialized in lower limb rehabilitation. The paper includes mechanical design, sensorics, trajectory generation and control strategy. In this sense, it is a very complete work. However, there are certain aspects that need to be improved. Specifically, the main weaknesses of the article are: i) It does not justify why it does not affect the dynamics of the human body for the control strategy design.
Answer: The dynamics of the human in the control strategy was not affected due to:
“a dynamical model of the human is not obtained because we are performing only passive rehabilitation. That means that the patient must not make any effort during the performance of the exoskeleton movement routine. If for some reason the human generates any movement or effort, these are considered as disturbances and the proposed control is capable of rejecting them.”.
This justification was added in the Introduction section (line 100-105).
Reviewer 3: ii) It does not detail how to estimate, with an algorithm, the intention of movement and how this becomes the control reference;
Answer: We added Figure 16 in section 7.3 in order to explain the classification control technique.
In addition, to better explain the classification control technique, the following paragraph was added (line 328-332):
“In this section the control approach is described. Let us remember that: first, the human intention is detected using electromyography and baropodometry, then the system chooses which task will be developed (Sitting to standing, Standing to sitting or Sitting to walk), and selects one of the desired trajectories qdSTS, qdSTA or qdSW respectively, which were obtained by video analysis and are already programmed in the exoskeleton's computer.”
Reviewer 3: iii) It does not study the trajectories properties for following a dynamics of both speed and acceleration
Answer: We added Table 2, in which the upper bounds of the desired trajectories and velocities are presented. Furthermore, properties of the said signals were added in the text (line 233-235).
Reviewer 3: iv) It does not present any control specification for the adjustment of the controller.
Answer: A description to explain the adjustment of the controller was added in the paragraph: “and has the advantage that, to adjust the gains, the only condition is that $K_1$ must be positive and $K_2 >I$” (line 441-442).
Reviewer 3: Now, I detail the issues section by section. 1. Introduction
It mixes the state of the art with the description of the exoskeleton they have developed. This section should be devoted only to the state of the art and highlight the new contributions of the work.
Answer: The introduction chapter was modified to separate the part of the state of the art and the part of the contribution of the work.
Reviewer 3: Also, figure 1 does not present the same exoskeleton configuration as described later. It is confusing.
Answer: To clarify the Figure 1, an explanation was added in the Introduction section, line 80-87.
Reviewer 3: 2. Mechanical construction.
This section should include a complete description of the developed exoskeleton. It is also convenient to include a control architecture and its description to link all the elements that will be described in the following sections.
Answer: The Mechanical Construction Section was renamed as "Platform Development" to include a description of the elements that are presented in the other sections.
Reviewer 3: 3. Mathematical model.
The first question to be answered is why the dynamics of the person who will use the exoskeleton is not taken into account, and only the dynamics of the exoskeleton is taken into account. By methodology, before describing the dynamic model, it is convenient to present the direct kinematic model and the Jacobian model before approaching the calculation of the dynamic model. The construction of the dynamic model considers the kinetic and potential energy of the mechanism, making the calculations in formulae (3) and (4) redundant. It is sufficient to include the actuator with the elastic element in the dynamic model. Furthermore, the calculations do not include the gearbox, the effect of which is to make the system linear and thus convert it into a linear MIMO system. Thus, the stability study and the design of the controller are much simpler. Everything described in subsection 3.1.1. is redundant with respect to the calculations performed to build the dynamic model of the mechanism. With respect to the generation of the trajectories, the dynamic model is not used to constrain them in velocity and acceleration. It is important to take this into account in order not to saturate the actuators or to establish a comfortable dynamic of movement for the patient. The calculation of the trajectory generation is disconnected from the dynamic model. Another important issue is that the dynamic model is not validated. Are the parameters of the dynamic model correct? Has an estimation procedure been performed to calculate the parameters of the dynamic model?
Answer: The dynamics of the human in the control strategy was not affected due to:
“a dynamical model of the human is not obtained because we are performing only passive rehabilitation. That means that the patient must not make any effort during the performance of the exoskeleton movement routine. If for some reason the human generates any movement or effort, these are considered as disturbances and the proposed control is capable of rejecting them.”.
This justification was added in the Introduction section (line 100-105).
Reviewer 3: 4. Sensors.
This section should be linked to section 5, dedicated to intention detection.
Answer: Section 5 (Human intention) is linked to section 4 (Sensors), leaving only the section 4 renamed Human Intention Detection.
Reviewer 3: It should also include the specific algorithm used to convert this intention of movements into commands for trajectory generation or control.
Answer: We added Figure 16 in section 7.3 in order to explain the classification control technique.
In addition, to better explain the classification control technique, the following paragraph was added (line 328-332):
“In this section the control approach is described. Let us remember that: first, the human intention is detected using electromyography and baropodometry, then the system chooses which task will be developed (Sitting to standing, Standing to sitting or Sitting to walk), and selects one of the desired trajectories qdSTS, qdSTA or qdSW respectively, which were obtained by video analysis and are already programmed in the exoskeleton's computer.”
Reviewer 3: 5. Human intention.
This requires further formalization and linking to the previous section. It does not detail anything about the filtering of the signals or the classification algorithm to be used to detect the intention. A sensory system architecture is also needed.
Answer: The analysis of the EMG method was improved by addressing the following points:
In the Detection of human intention Section we added: “Then, the used sampling frequency is of 1 kHz. The signals are filtered using a band-pass filter type butter-worth of fourth order from 10 to 250 Hz to have better information.” (line 278-279).
Reviewer 3: 6. Feedback control.
The authors are strongly based on a control strategy and stability analysis already published in "Modeling and control of robot manipulators" by L. Sciavicco and B. Siciliano, ISBN 0-07-057217-8. Siciliano, ISBN 0-07-057217-8. So, this section can be summarized and refer to this book. The point is that the dynamic model, when converted into a MIMO system, can simplify the stability analysis. Other unanswered questions: What are the control specifications used to calculate the controller gains?, How do the dynamics of the human body affect the controller performance), Is it necessary to consider a shared control scheme?.
Answer: We summarized the control algorithm and included the cited reference: [24] Pham, D. Modeling and control of robot manipulators by L Sciavicco and B Siciliano. Robotica. 1998, 16 6. ISBN 0–07–057217–8 521
DOI:10.1017/S0263574798220856.
Reviewer 3: 7. Numerical results.
This section and section 8 can be merged.
Answer: Section 7 and Section 8 were merged and improved to add a better analysis.
Reviewer 3: How the signals are used to plan the exoskeleton trajectories?.
Answer: The trajectories were obtained through video analysis, to clarify how the signals are used to plan the exoskeleton trajectories an explanation was added in the Feedback control Section line 328-332
Reviewer 3: In figure 10, the units of the "Y" axis are missing.
Answer: Figure 10 and the other Figures were revised to correctly add the units and legends of the graphs.
Reviewer 3: Section 7.2. should be part of section 5.
Answer: Subsection 7.2 was moved to become part of Section 5, considering that sections 4 and 5 were already linked by a previous suggestion, then, the Subsection 4.1.1 (Signal threshold detection) is now part of Section 4 renamed Detection of human intention.
Reviewer 3: 8. Real-time experimental results.
The conditions under which each experiment is performed should be specified. What is the weight, height of the patient using it?
Answer: In the experimental results section, we specified the parameters in each experiment and the conditions. The following paragraph was added: “The human intention identification system was proved with a healthy female subject of 27 years with a weight of 55 kg and a height of 1.6 m.”
Reviewer 3: Does this affect the performance of the controller?
Answer: We tried to follow the trajectory references in a smooth way in order to prevent injury, this action produces a small drift between the references trajectory and final joints.

Round 2
Reviewer 1 Report
The thesis is well organized.
I would like to increase the visibility, such as the size of the text in the picture.
Author Response
Once again, we thank you for your comments, which were very useful in the review of this paper.
Reviwer 1: I would like to increase the visibility, such as the size of the text in the picture.
Answer: The text in the figures was increased

Reviewer 2 Report
Thank you for your response to my comments.
Please confirm one point below.
In response to my comment where I asked for the filter to be listed, the manuscript states 10-250 Hz, but the author's response states 10-500 Hz.
Please confirm which is correct.
Author Response
Once again, we thank you for your comments and observations that were very useful.
Reviewer 2:
In response to my comment where I asked for the filter to be listed, the manuscript states 10-250 Hz, but the author's response states 10-500 Hz.
Please confirm which is correct.
Answer:
The correct answer about the filter is 10-250 Hz, as in the manuscript. The paper is attached with the last changes and observations highlighted in yellow

Reviewer 3 Report
I have reviewed the answers to the review and the modifications made to the paper. I am satisfied with all of them, but I have some doubts about one issue that should be better explained by the authors. In particular, it is about how the dynamics of the human body affects the control, even if the rehabilitation is passive.
The control strategy is designed based on the dynamics of the exoskeleton that takes into account the mass of each of its elements. However, when a person uses the exoskeleton, his or her mass distribution is linked to the device and changes the inertias, centres of mass, inertial tensors etc. In this way, the controller is working with wrong parameters of the dynamic model. The controller only knows the dynamics of the exoskeleton itself, but not of the exoskeleton-human combined dynamic.
There must be some mechanism to compensate this effect in such a way the controller works properly. There are elements of the prototype shown in Figure 1 that are not taken into account in Figure 4 that cancel the human mass contribution to the exoskeleton dynamics. Indeed, in the walking task, the effect of the mass added by the human may be negligible, but not in the sitting and standing task. The weight of the human adds gravitational and inertial torques that are not considered when designing the controller.
The authors need to justify why the mass of the human when using the exoskeleton, which adds gravitational and inertial torques, is negligible in the design of the
controller design. Likewise, they should also justify how the described problem affects the planning of the trajectories used in each of these rehabilitation tasks.
Author Response
Once again, we thank you for your comments and observations that were very useful.
Reviewer 3:
I have reviewed the answers to the review and the modifications made to the paper. I am satisfied with all of them, but I have some doubts about one issue that should be better explained by the authors. In particular, it is about how the dynamics of the human body affects the control, even if the rehabilitation is passive.
The control strategy is designed based on the dynamics of the exoskeleton that takes into account the mass of each of its elements. However, when a person uses the exoskeleton, his or her mass distribution is linked to the device and changes the inertias, centres of mass, inertial tensors etc. In this way, the controller is working with wrong parameters of the dynamic model. The controller only knows the dynamics of the exoskeleton itself, but not of the exoskeleton-human combined dynamic.
There must be some mechanism to compensate this effect in such a way the controller works properly. There are elements of the prototype shown in Figure 1 that are not taken into account in Figure 4 that cancel the human mass contribution to the exoskeleton dynamics. Indeed, in the walking task, the effect of the mass added by the human may be negligible, but not in the sitting and standing task. The weight of the human adds gravitational and inertial torques that are not considered when designing the controller.
The authors need to justify why the mass of the human when using the exoskeleton, which adds gravitational and inertial torques, is negligible in the design of the
controller design. Likewise, they should also justify how the described problem affects the planning of the trajectories used in each of these rehabilitation tasks.
Answer:
The proposed control law is based on the exoskeleton dynamic model, this exoskeleton mobilizes the hip and knee joints which is seen in figure 1.
Additionally, the complete rehabilitation system has integrated the lifting system and the treadmill. Both mechanisms, together, support the entire weight of the patient, including that of the legs as seen in figure 4. Due to these systems, the effect of the mass added by the patient to the exoskeleton can be neglected, not only during the walking routine, but even in the movements of sitting and standing, since it is the lifting system that, in this case, compensates the effect of mass added by the patient.
In fact, when a rehabilitation is performed, the most convenient is to adjust the control gains, this action is very simple only tunning the gains in the sense of closed-loop stability.
In the paper, a restructuring of the conclusions underlined in yellow was made, which contains an explanation to clarify your observation in the paper. Line 441-447
